# SARS-CoV-2 detection in setting of viral swabs scarcity: Are MRSA swabs and viral swabs equivalent?

Daniel G. Federman[1,2]*, Shaili Gupta[1,2], Gary Stack[1,2], Sheldon M. Campbell[1,2], David R. Peaper[1,2], Louise M. Dembry[1,2], Ann Fisher[1,2], Asim F. Tarabar[1,2], Michael Kozal[1,2], Christopher B. Ruser[1,2]

1 VA Connecticut Healthcare System, West Haven, CT, United States of America, 2 Yale University School of Medicine, New Haven, CT, United States of America

* Daniel.federman@va.gov

**Data Availability Statement:** all relevant data are within the manuscript and its supporting information files.

## Abstract

### Background

The global pandemic of Severe Acute Respiratory Syndrome-Related Coronavirus 2 (SARS-CoV2) has resulted in unprecedented challenges for healthcare systems. One barrier to widespread testing has been a paucity of traditional respiratory viral swab collection kits relative to the demand. Whether other sample collection kits, such as widely available MRSA nasal swabs can be used to detect SARS-CoV-2 is unknown.

### Methods

We compared simultaneous nasal MRSA swabs (COPAN ESwabs ® 480C flocked nasal swab in 1mL of liquid Amies medium) and virals wabs (BD H192(07) flexible mini-tip flocked nasopharyngeal swabs in 3mL Universal Transport Medium) for SARS-CoV-2 PCR testing using Simplexa COVID-19 Direct assay on patients over a 4-day period. When the results were discordant, the viral swab sample was run again on the Cepheid Xpert Xpress ® SARS-CoV-2 assay.

### Results

Of the 81 included samples, there were 19 positives and 62 negatives in viral media and 18 positives and 63 negative in the MRSA swabs. Amongst all included samples, there was concordance between the COPAN ESwabs ® 480C and the viral swabs in 78 (96.3%).

### Conclusion

We found a high rate of concordance in test results between COPAN ESwabs ® 480C in Amies solution and BD H192(07) nasopharyngeal swabs in in 3 mL of Universal Viral Transport medium viral media. Clinicians and laboratories should feel better informed and assured using COPAN ESwabs ® 480C to help in the diagnosis of COVID-19.

**Funding:** There was no funding involved in this manuscript.

**Competing interests:** The authors have declared that no competing interests exist.

## Introduction

As early as December 2019, a cluster of patients with pneumonia of unknown etiology was identified in Wuhan, China [1] and shortly thereafter, the causative agent was identified as a novel betacoronavirus, the severe acute respiratory syndrome-related coronavirus-2 (SARS-CoV-2) [2]. The spectrum of disease severity ranges from asymptomatic disease to severe pulmonary disease including acute respiratory distress syndrome, multisystem organ failure, and death [1]. Genetic sequencing performed early in the course of the outbreak led to the development of diagnostic testing [3].

With testing, SARS-CoV-2 was noted to be responsible for a global pandemic, and as of April 27, 2020, a total of nearly 3 million cases worldwide have been diagnosed, and at least 207,000 have died from COVID-19 (coronavirus disease 2019) [4]. Many experts have advocated for widespread testing. Due to the rapidity of and extensive spread of the infection however, many countries, including the United States, were ill-prepared to employ widespread testing as an effective public health tool [5]. Limitations related to the availability of testing were complex, and occurred on many levels: first, various assays had to be developed and deployed, second, physical testing sites (including drive-through) had to be operationalized, and lastly supplies of ancillary equipment had to be obtained. Key to ancillary equipment was adequate and sustained supply of viral sample collection kits. Many healthcare systems that have overcome the first two of these barriers have been faced with unexpected shortages of swabs and transport media, compounded by slow or non-existent replacements and unreliable supply chains.

As with creative interventions to re-purpose personal protective equipment (PPE), healthcare systems have looked for ways to overcome barriers to testing. Interim guidelines issued by the CDC on April 14, 2020 clarified the allowance of other swab types with guidance on specimen collection and transport. In these guidelines, both FDA and CDC allowed for expansion of the specimen types and swab/transport media to accommodate demand for more testing. While nasopharyngeal specimens obtained with a "mini-tip" viral swab and transported in viral transport medium is still the preferred choice for initial testing, acceptable alternatives include nasopharyngeal aspirates and nasal washes as well as swabs of oropharynx, anterior nares, and nasal mid-turbinates. The latter two are only appropriate in symptomatic patients and both nares must be sampled. The anterior nares specimen should be collected using a spun polyester or flocked COPAN ESwabs® used for MRSA detection. These interim guidelines allow for anterior nares and mid-turbinate specimens to be transported in viral transport medium, Amies transport medium, or sterile saline. There is no data, however on the effect on diagnostic accuracy of SARS-CoV-2 tests based on these variable swabs and transport media used for specimen collection from sites other than the nasopharynx.

Given the wide availability of specimen collection kits used for MRSA testing in our facility, we sought to determine whether nasal MRSA swabs (flocked polyester COPAN ESwab® 480C) and their transport medium (1mL of liquid Amies medium) can be used to detect SARS-CoV-2 when viral swabs are not available. We therefore sought to assess the concordance between the test results using these two different collection kits. This of paramount importance at this critical time.

## Methods

Nasal/nasopharyngeal swabs samples were obtained at the VA Connecticut Healthcare System from either symptomatic patients or patients with a high-risk exposure to COVID-19 presenting to a pre-scheduled "drive-through" outpatient testing center, its emergency department, as well as the inpatient and medical intensive care COVID-19 units over a four-day period.

Hospitalized inpatients on the inpatient unit and medical intensive care unit had been initially diagnosed with SARS-CoV-2 from one day to 24 days prior to re-testing.

Each patient underwent concomitant testing with both a "viral swab" in Becton, Dickinson and Company H192(07) flexible fine-tip flocked nasopharyngeal swab, with 3 mL of Universal Viral Transport medium (Becton Dickinson H192(07) and an "MRSA swab" COPAN ESwab ® 480C with a flocked polyester nasal swab in 1mL of liquid Amies solution. Trained nurses or physicians collected the specimens. The viral swab specimens were nasopharyngeal; each MRSA swab specimen was collected by accessing a single nare, and extending the swab as far into the mid-turbinate area as the thickness of the swab would allow. After collection, specimens were stored at room temperature until received by the lab and subsequently at -20 degrees Celsius. All specimens were tested within 24 hours of collection. We found that the COPAN ESwab® was not compatible with the Roche Cobas 6800 machine due to the high concentration of mucus and particulate matter that is incompatible with that instrument's sampling apparatus. We therefore tested both samples with the Simplexa ® COVID-19 Direct Kit (DiaSorin). This real-time RT-PCR assay used manufacturer-provided fluorescent probes with corresponding forward and reverse primers to target two regions of the SARS-CoV-2 genome, the S gene and ORF1ab [6]. When the MRSA and viral swabs were discordant, the viral swab was run on the Xpert® Xpress SARS-CoV-2 (Cepheid). The protocol was considered a quality improvement project and granted exemption by the VA Connecticut Research Office and deemed that review by the VA Connecticut Investigational Review Board (IRB)or R&D Committee was not needed since the objective was optimization of testing supplies. The VA Connecticut IRB provided written confirmation that IRB approval was not necessary. A waiver of informed consent was obtained from its Human Studies Subcommittee.

## Results

We sampled 81 unique patients, of which the viral swabs were positive for SARS-CoV-2 PCR on the DiaSorin test in 19. Of these samples, there were 19 positives and 62 negatives in viral media and 18 positives and 63 negatives in the MRSA collection kits (Fig 1). Amongst all included samples, there was concordance between the COPAN ESwabs ® 480C and the viral

|  |  | MRSA Swabs | | |
|---|---|---|---|---|
|  |  | + | - | Total |
| Viral Swabs | + | 17 | 2 | 19 |
|  | - | 1 | 61 | 62 |
|  | Total | 18 | 63 | 81 |

Fig 1. Test esults on the Simplexa ® COVID-19 Direct Kit (DiaSorin) comparing viral swabs and MRSA swabs.

swabs in 78 (96.3%). In one patient the viral swab was negative and the COPAN ESwabs [R] positive and in two patients the viral swab was positive and the COPAN ESwabs [R] negative. When the results were discordant, the viral swab was re-tested with the Cepheid assay. When this was done, in two cases, the COPAN ESwab [R] 480C result was concordant with the Cepheid platform. Concordance was achieved in one case because a negative result with the viral swab on the DiaSorin assay was presumptive positive on the Cepheid, and in the other case because a negative result on the DiaSorin with the COPAN ESwab [R] was positive with Cepheid.

## Discussion

SARS-CoV-2 infection has overwhelmed healthcare systems internationally and caused unparalleled morbidity and mortality. Efforts to mitigate this infection focus on social distancing, widespread testing, home isolation, and timely implementation of precautions for hospitalized patients. In order to meet the need for diagnosis of COVID-19, several testing platforms have been approved by FDA, and several more may become available in the near future. Currently available reverse-transcriptase PCR tests that have been granted emergency use authorization from the FDA target specific genes of SARS-CoV2, and they are thought to provide relatively comparable and accurate results. However, like most other healthcare systems across the US, our healthcare system is experiencing a shortage of viral respiratory swabs and viral transport medium. While CDC and FDA have modified their sample collection guidelines to include specimens from nares and mid-turbinate using polyester flocked swabs transported in a variety of media, the effect of these on the diagnostic accuracy of the RT-PCR tests is not known. The concept of assessment of validity of SARS-CoV2 testing using COPAN ESwabs [R] 480C was therefore born out of necessity.

Our primary concerns were the anticipated mismatch in the site of sampling, and the concern about RNA-survival in the Amies solution. The COPAN Eswab [R] is thicker than the viral respiratory swab because the targeted site of sampling for MRSA is the anterior nares. This is in contrast to the respiratory viral swab, which has a flexible mini-tip and is designed to sample beyond the turbinates, to the nasopharyngeal wall, where the yield of respiratory viruses is typically higher since the local ciliated epithelial cells support viral replication, in contrast to the squamous epithelium of the anterior nares. The yield of SARS-CoV2 from anterior nares sampling has been expected to be inadequate analogous to influenza testing, but data are limited and contradictory [7].

Similarly, the Amies solution that accompanies the swabs for bacterial preservation does not contain any antimicrobial agent in order to prevent any compromise in the yield of bacterial genetic material. Instead, it contains salts to provide essential ions, inorganic phosphate to provide buffering, and sodium thioglycolate to provide reduced environment, meant to maintain viability of bacteria during transport. This, however, can be an undesirable medium for viral transport for that very reason. Abundant bacteria, when present, may commence a destructive effect on viral RNA almost immediately after sample collection. Time from specimen collection to receipt laboratory is therefore of utmost importance for such substitutions. To minimize such concerns we ensured that our samples were transported to the laboratory expeditiously, then frozen at -20 degrees Celsius to abort any bacterial activity. We find the high concordance rate between the swabs reassuring for reproducibility of this particular substitution.

Concerns regarding SARS-CoV-2 RNA stability in liquid Amies, however, were recently addressed by Rogers et al [8]. They found that SARS-CoV-2 RNA, when spiked into liquid Amies specimen remnants from COVID-19 negative patients, was stable for at least seven days

at room temperature and at least 14 days when refrigerated. They did note an increase in cycle threshold values of about 2–3 after two weeks of frozen storage, but those changes did not alter result interpretations. Taken together, their results with ours support the suitability of ESwabs with liquid Amies for COVID-19 testing.

It is interesting to note that in two of the three cases when the results of the viral swab and COPAN ESwab ® were discordant on the DiaSorin platform, the Cepheid platform gave concordant results. In both cases the concordance was achieved because the Cepheid obtained a positive result where the Diasorin had obtained a negative. This is perhaps related to the fact that the Cepheid interprets results as positive up to a cycle threshold value of 44, whereas the Diasorin's cut-off for positives is 40. A recent news report of unpublished data suggests that the DiaSorin test has a false-negative rate of nearly 11% and Cepheid a false negative result in 1.8% [9].

An unexpected finding was that the Ct values for detection of the ORF1ab sequence were significantly lower overall with the COPAN ESwab® than for the viral swab. Also, if it weren't for two outliers, the same would have been true for the Ct values for S gene detection. We speculate that this might be related to the fact that the tip of the COPAN ESwab® is larger than that of the viral swab, and might collect a larger sample with a larger viral load. This would be consistent with our anecdotal observation that specimens collected with the COPAN ESwab® contained more mucus. This finding could also be related to the fact that viral RNA concentration may be higher in the COPAN ESwab® specimens because the volume of liquid Amies transport medium is one third the volume of UTM used with the viral swab (1 mL versus 3 mL). At a minimum, we can conclude that the COPAN ESwab® with liquid Amies transport medium appears as sensitive for detecting SARS-CoV-2 RNA as the viral swab in UTM.

There are several limitations to our study. Our sample size is both small and from one center, and needs to be confirmed in future larger studies including multiple centers with multiple technicians obtaining samples. Furthermore, we used two proprietary testing formulations. Whether this is generalizable to other formulations of viral and non-viral sampling systems is not known. Ideally, we would have liked to test each sample on multiple platforms looking for concordance between several testing modalities, as it is conceivable that inhibitory substances may be present in nasal materials or different transport media. However, due to the ongoing relative shortage of available reagents, we could not justify this for this quality improvement project.

In addition to its tremendous physical and emotional toll, infection caused by SARS-CoV-2 has stressed supply chains. While some may have used typical MRSA swabs when there have been shortages of viral swabs, heretofore there has been little evidence of their testing characteristics. We found a high rate of concordance in test results between COPAN ESwabs ® 480C in Amies solution and swabs in viral media. Clinicians and laboratories should feel better informed and assured that using COPAN ESwabs ® 480C to help in the diagnosis of COVID-19 is acceptable in resource-limited settings.

## Supporting information

**S1 Data.**
(XLSX)

## Author Contributions

**Conceptualization:** Shaili Gupta, Gary Stack, Michael Kozal.

**Data curation:** Daniel G. Federman, Gary Stack, Sheldon M. Campbell, Asim F. Tarabar, Christopher B. Ruser.

**Formal analysis:** Daniel G. Federman, Gary Stack.

**Investigation:** Daniel G. Federman, Shaili Gupta, Gary Stack, Sheldon M. Campbell, David R. Peaper, Louise M. Dembry, Ann Fisher.

**Methodology:** Daniel G. Federman, Sheldon M. Campbell, David R. Peaper.

**Project administration:** Louise M. Dembry, Ann Fisher, Michael Kozal.

**Supervision:** Daniel G. Federman, Michael Kozal, Christopher B. Ruser.

**Validation:** David R. Peaper.

**Writing – original draft:** Daniel G. Federman.

**Writing – review & editing:** Gary Stack, Sheldon M. Campbell, David R. Peaper, Louise M. Dembry, Ann Fisher, Asim F. Tarabar, Michael Kozal, Christopher B. Ruser.

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
