## [Decision Letter · Decision Letter 0]

10 Jun 2020

PONE-D-20-14345

SARS-CoV-2 detection in setting of viral swabs scarcity: are MRSA swabs and viral swabs equivalent?

PLOS ONE

Dear Dr. Federman,

Thank you for submitting your manuscript to PLOS ONE. After careful consideration, we feel that it has merit but does not fully meet PLOS ONE’s publication criteria as it currently stands. Therefore, we invite you to submit a revised version of the manuscript that addresses the points raised during the review process.

As mentioned by Reviewer#1, this study was done using limited number of samples from one location.  This should be mentioned in the text as a limitation of this study. All other comments of all reviewers should be addressed on the revised manuscript in addition to the rebuttal letter.

We look forward to receiving your revised manuscript.

Kind regards,

Selvakumar Subbian, Ph.D.

Academic Editor

PLOS ONE

Journal Requirements:

2. Thank you for stating in your ethics statement

"The protocol was deemed a quality improvement project and granted exemption by the VA Connecticut Research Office and deemed that review by the VA Connecticut Investigational Review Board and R and D Committee was not needed."

Please clarify whether the  VA Connecticut Research Office is your ethics committee.

Please also clarify whether informed consent was specifically waived by your ethics committee.

If not, please discuss whether all data were fully anonymized before you accessed them or whether written informed consent was obtained from patients.

4. We noted in your submission details that a portion of your manuscript may have been presented or published elsewhere:

'The entire manuscript was submitted simultaneous with MedRxiv and published there. There was supposed to be simultaneous submission to your journal but there was a problem with submission (linking of my account with the account of a Dr. David Feillin- your office knows about this)'

Please clarify whether this publication was peer-reviewed and formally published.

If this work was previously peer-reviewed and published, in the cover letter please provide the reason that this work does not constitute dual publication and should be included in the current manuscript.

6. Please ensure that you refer to Figure 1 in your text as, if accepted, production will need this reference to link the reader to the figure.

Reviewers' comments:

Reviewer's Responses to Questions

**Comments to the Author**

1. Is the manuscript technically sound, and do the data support the conclusions?

Reviewer #1: Partly

Reviewer #2: Yes

2. Has the statistical analysis been performed appropriately and rigorously? 

Reviewer #1: No

Reviewer #2: N/A

3. Have the authors made all data underlying the findings in their manuscript fully available?

Reviewer #1: Yes

Reviewer #2: Yes

4. Is the manuscript presented in an intelligible fashion and written in standard English?

Reviewer #1: Yes

Reviewer #2: Yes

5. Review Comments to the Author

Reviewer #1: The manuscript entitled SARS-CoV-2 detection in setting of viral swab scarcity: are MRSA swabs and viral swabs equivalent? describes the use of MRSA swabs with Amies media and Viral swab with universal transport media for sample collection from SARS-CoV-2 infected patients. Authors indicated that the MRSA swabs yields the comparable results as that of viral swabs. However, this manuscript failed to address the following;

1. Sample size is less

2. Sampling was done at only one center; sampling in different screening center by different technicians will require to exclude personal variation in sample collection.

3. As the Amies media is not known as medium of choice for sample collection for viral diagnosis and it is expected that the sample will be stored for varying time period at room temperature prior to transport to lab to store at -20C; the duration of stability of RNA in the Amies media must be tested prior to recommending them for SARS-CoV-2 sample collection.

4. References not included for several sentences in the introduction; For example- "The spectrum of disease severity ranges from asymptomatic disease to severe pulmonary disease including acute respiratory distress syndrome, multisystem organ failure, and death. Genetic sequencing performed early in the course of the outbreak led to the

development of diagnostic testing".....include references.

5. The following sentence is not clear and need to be re-written : We sampled 81 unique patients, of which the viral swabs were positive on the DiaSorin test in 19.

6. It is needless to mention "Additionally, while our study was conducted only in Veterans, we do not believe there

should be a difference in test results between Veterans and non-Veterans for the positive or

negative predictive value of testing for SARS-CoV-2 infection"

7. Under methods, the sentence "virals wabs (BD H192(07) flexible mini-tip flocked " need to be written as "viral swabs (BD H192(07) flexible mini-tip flocked"

8. The very short and incomplete sentences [ for example "The viral swab specimens were nasopharyngeal" needs to be merged with neighboring sentence or can be re-written appropriately.

Reviewer #2: The tremendous need for swab test justifies the work by Daniel et.al. and this would for sure be helpful in terms of increasing the capacity to test. The manuscript is technically sound and the data support the conclusions. Although the sample number is small, the importance of work makes it acceptable.

Percentage correlation of the positive and negative test of two different swabs together with results from Cepheid Xpert Xpress ® SARS-CoV-2 assay justify the conclusions with appropriate control.

Minor revisions: The authors should describe the primers used for the amplification of CoV2 RNA with details of the amplification conditions in the method section.

The manuscript is well written and acceptable with minor reversions.

6. PLOS authors have the option to publish the peer review history of their article (what does this mean?). If published, this will include your full peer review and any attached files.

Reviewer #1: No

Reviewer #2: Yes: Alok Choudhary

---

## [Decision Letter · Decision Letter 1]

22 Jul 2020

SARS-CoV-2 detection in setting of viral swabs scarcity: are MRSA swabs and viral swabs equivalent?

PONE-D-20-14345R1

Dear Dr. Federman,

We’re pleased to inform you that your manuscript has been judged scientifically suitable for publication and will be formally accepted for publication once it meets all outstanding technical requirements.

Kind regards,

Selvakumar Subbian, Ph.D.

Academic Editor

PLOS ONE

Additional Editor Comments (optional):

Reviewers' comments:

Reviewer's Responses to Questions

**Comments to the Author**

1. If the authors have adequately addressed your comments raised in a previous round of review and you feel that this manuscript is now acceptable for publication, you may indicate that here to bypass the “Comments to the Author” section, enter your conflict of interest statement in the “Confidential to Editor” section, and submit your "Accept" recommendation.

Reviewer #1: (No Response)

2. Is the manuscript technically sound, and do the data support the conclusions?

Reviewer #1: (No Response)

3. Has the statistical analysis been performed appropriately and rigorously? 

Reviewer #1: (No Response)

4. Have the authors made all data underlying the findings in their manuscript fully available?

Reviewer #1: (No Response)

5. Is the manuscript presented in an intelligible fashion and written in standard English?

Reviewer #1: (No Response)

6. Review Comments to the Author

Reviewer #1: Authors have addressed all the comments.

However, due to the small sample size tested for the evaluation of MRSA swabs, it will be more appropriate if the title of manuscript changed as "Pilot study on the applicability of MRSA swabs for SARS-CoV-2 sampling in setting of viral swabs scarcity".

7. PLOS authors have the option to publish the peer review history of their article (what does this mean?). If published, this will include your full peer review and any attached files.

Reviewer #1: **Yes: **Santhamani Ramasamy

---

## [Editor Report · Acceptance letter]

28 Jul 2020

PONE-D-20-14345R1 

SARS-CoV-2 detection in setting of viral swabs scarcity: are MRSA swabs and viral swabs equivalent? 

Dear Dr. Federman:

I'm pleased to inform you that your manuscript has been deemed suitable for publication in PLOS ONE. Congratulations! Your manuscript is now with our production department. 

Kind regards, 

on behalf of

Dr. Selvakumar Subbian 

Academic Editor

PLOS ONE